# The Role of Microbial Factors in Prostate Cancer Development—An Up-to-Date Review

**DOI:** 10.3390/jcm10204772

**Published:** 2021-10-18

**Authors:** Karolina Garbas, Piotr Zapała, Łukasz Zapała, Piotr Radziszewski

**Affiliations:** Department of General, Oncological and Functional Urology, Medical University of Warsaw, Poland Lindleya 4, 02-005 Warsaw, Poland; kgarbas98@gmail.com (K.G.); zapala.lukasz@gmail.com (Ł.Z.); pradziszewski@wum.edu.pl (P.R.)

**Keywords:** microbiome, microbiota, sexually transmitted infections, prostate cancer

## Abstract

Up-to-date studies emphasize the role of human urinary and intestinal microbiome in maintaining urogenital health. Both microbial flora and sexually transmitted pathogens may affect metabolic or immune mechanisms and consequently promote or inhibit prostate carcinogenesis. Hereby, we review the most current evidence regarding the microbial factors and their link to prostate cancer. We conducted a literature search up to December 2020. The microbial impact on prostate cancer initiation and progression is complex. The proposed mechanisms of action include induction of chronic inflammatory microenvironment (*Propionibacterium* spp., sexually-transmitted pathogens) and direct dysregulation of cell cycle (*Helicobacter pylori*, Kaposi’s sarcoma-associated herpesvirus- KSHV, human papilloma virus 18- HPV18). Suppression of immune cell expression and downregulating immune-associated genes are also observed (*Gardnerella vaginalis*). Additionally, the impact of the gut microbiome proved relevant in promoting tumorigenesis (*Bacteroides massiliensis*). Nevertheless, certain microbes appear to possess anti-tumor properties (*Listeria monocytogenes*, *Pseudomonas* spp.), such as triggering a robust immune response and apoptotic cancer cell death. The role of microbial factors in prostate cancer development is an emerging field that merits further studies. In the future, translating microbial research into clinical action may prove helpful in predicting diagnosis and potential outcomes of the disease.

## 1. Introduction

Prostate cancer is the most common malignancy in men and the second leading cause of cancer-related deaths among males in the United States [1]. In Europe, prostate cancer accounted for approximately 23% of all new malignancies diagnosed in male patients in 2020 [2]. The highest incidence rates were found in northwestern Europe, with Ireland in the lead [3]. Age and ethnicity are established risk factors, whereas postulated factors include lifestyle as well as environmental and occupational exposures [4]. However, none of the aforementioned factors has proved to be dominant in pathogenesis of prostate cancer and direct inductors of oncogenesis are yet to be discovered. Thus, the role of microbial factors, including human urinary and gut microbiome, is gaining attention [5]. The human microbiota can be defined as microorganisms, such as bacteria, archaea, fungi, and protozoa, living in the epithelial barrier surfaces of the body [6]. The term microbiome is used as a reference to the habitat as a whole, incorporating both the biotic and abiotic factors [7]. The relationship between these active organisms and urogenital health is yet to be elucidated. Symbiotic equilibrium between the host and its microbiota is expected. Nevertheless, various stressors, such as drugs, environmental factors, and exogenous pathogenic bacteria, can lead to dysbiosis and, eventually, promote many diseases, including cancer [8]. The microbiome possibly affects different stages of oncogenesis, from initiation to progression, as well as treatment efficacy. This influence could be through direct interactions at the site of carcinogenesis and indirect mechanisms, such as regulating the immune system and metabolic changes [9].

Apart from the human microbiome, sexually transmitted infections (STIs) have also been suggested to increase prostate cancer risk [10]. Exogenous pathogens may cause chronic inflammation within the prostatic tissue, thus provoking uncontrolled cell proliferation and ultimately induce carcinogenesis [11]. It is speculated that history of multiple episodes of STIs or untreated infections could result in an especially higher risk of prostate cancer development [12]. This review explores the connections between the microbial factors and prostate cancer, with a specific focus on human microbiome, as well as exogenous, sexually transmitted pathogens.

## 2. Microbiome Detection Techniques—How Did We Get Here?

The issue of urinary microbiome and its impact on various aspects of urogenital health has only recently been raised. Although urine was traditionally thought to be sterile, studies have challenged this decades-old dogma and confirmed that the human urinary tract harbors a resident microbiome. An original study by Anderson et al. proved the presence of viable, but nonculturable, bacteria in human clean-catch and mouse bladder-isolated urine specimens [13]. Nevertheless, it was not until the introduction of advanced molecular-based detection techniques that a resident urinary microbial niche could be thoroughly studied [13,14,15]. These highly specific modern methods allow for more accurate microbes identification, in contrast to traditional microscopic detection or cultivation [14]. After the discovery of urinary microbiome, expressed prostatic secretions [16] and cancer tissue [17,18,19] also became an object of interest for possible microbial presence.

Currently, the most widespread and highly available technique, used for the identification, classification, and quantitation of bacteria, involves polymerase chain reaction (PCR) and 16S rRNA gene sequencing [14,17]. Other variations of PCR, introduced subsequently, have also gained traction as helpful detection tools. These include a real-time PCR, particularly quantitative real-time PCR [15] and PCR-denaturing gradient gel electrophoresis (PCR-DGGE) [16].

While 16S rRNA gene sequencing has been a mainstay of sequence-based bacterial analysis, identification of viruses requires metagenomic sequencing due to their lack of the aforementioned phylogenetic marker [17]. Recent introduction of whole-genome sequencing and bioinformatic data-analyzing platforms enables even more precise and unbiased detection of resident microbial DNA [20,21]. The most recent and still advancing sequencing method, that offers a long read size, is the massive ultradeep pyrosequencing. Due to high costs and the need of confident data treatment and analysis tools, it is not yet commonly available [22].

## 3. Can Microbiota Maintaining Chronic Inflammation Be Attributed to Prostate Cancer Development?

The urinary tract is considered the main infiltration route for potentially pathogenic bacteria which might trigger neoplastic transformation by induction of chronic inflammatory microenvironment. Several bacteria species, including Propionibacterium acnes [21,22,23] and Mycoplasma genitalium [24], were initially postulated as driving tumorigenesis in an inflammatory-based mechanism. Microbiome profiling of prostate cancer and healthy tissues revealed more abundant Propionibacterium spp. and Staphylococcus spp. representation in the tumor and peri-tumor tissues (*p* < 0.05) [22]. Whole genome sequencing data introduced Proteobacteria as dominating microbes, enriching prostate tumor core [21].

Continuous proliferation of the aforementioned pathogenic bacteria might eventually change or outcompete natural microbiome of the prostatic tissue. This may lead to chronic inflammation and immune imbalance. When stimulated by bacterial components, phagocytes (mainly macrophages) release reactive oxygen species and reactive nitrogen species. These factors may directly damage DNA and cause genetic instability [11]. Thus, induced oxidative stress and the consequent cellular damage can stimulate proliferation of atrophic luminal epithelial cells, creating regions known as proliferative inflammatory atrophy (PIA), i.e., precursors of prostate malignancies [25].

Indeed, analysis of urinary microbiome proved that the urine of patients with biopsy-proven prostate cancer contained differentially abundant uropathogens, significantly less prevalent in control samples, such as *Anaerococcus lactolyticus*, *Varibaculum cambriense*, and *Propionimicrobium lymphophilum*. *Actinobaculum schaalii*, an emerging, yet underestimated uropathogen and *Ureaplasma* were both detected in men with cancer, for that matter [15]. Almost all of these species have been implicated as causative agents in urogenital infections, including prostatitis and urinary tract infections, which strongly suggest an inflammatory process and its possible impact on carcinogenesis.

Another study indicated that the urinary microbiota compositions in the expressed prostatic secretions (EPS) of prostate cancer samples were similar to those seen in benign prostate hyperplasia (BPH). Nevertheless, significant differences in quantities of specific strains of bacteria were found, including *Bacteroidetes*, *Sphingomonas*, *Firmicutes*, and *Ochrobactrum*.

Compared to patients with BPH, those diagnosed with prostate cancer had higher incidence of occult blood and turbidity, along with a higher quantity of white blood count and bacteria in the urine. This particular finding suggests that these patients might experience concurrent infection and inflammation of the prostate or urinary tract. Last but not least, the abundant presence of *Ochrobactrum*, an opportunistic pathogen, suggests immune dysfunction in prostate cancer patients and might be connected to cancer (stage) progression [16].

A precise mechanism of affecting prostatic tissue by herein indicated bacteria is yet unknown, and further studies are needed. A significant part of available data do not report any specific microbial species in the cancer group or controls, emphasizing mainly quantitative differences [26,27]. Thus, it is the overall microbial pattern, rather than a single species, that might play a crucial role in proliferative inflammation.

## 4. Single Species over Quantity—Can Microbiome Directly Stimulate Oncogenesis?

Numerous studies report a lower overall microbial diversity in the urine of prostate cancer patients, which stays in contraposition to the chronic inflammation hypothesis. Having analyzed prostate cancer tissue samples, an original study detected and compared microorganisms between patients of different ethnic origins. Twenty-three common bacterial genera were identified in the African and European-derived prostate tumor samples. The most abundant strains across all samples included Escherichia, Propionibacterium, and Pseudomonas [21], which stayed consistent with a complimentary study conducted in China [19]. *Bacteroides*, *Eubacterium*, *Parabacteroides*, and *Odoribacter* were yet exclusive to the African-derived tumor tissue. This could suggest the possibility of bacterially induced oncogenic transformation, which contributes to an aggressive disease presentation visible within the African population [21]. Surprisingly, no correlation was found between the presence of these bacteria and clinical presentation of the disease, i.e., either high-risk or low-risk cancer [21].

Contrarily, other research reports significantly higher microbial levels and even signatures of parasitic (such as *Plasmodium*) presence in prostate cancer tissue [17]. Nevertheless, the levels of microorganisms detected are still considered relatively low. Hence, the general conclusion against a meaningful replicative infection and a need for another potential mechanism of neoplastic initiation. Disturbing the cell cycle by integration of microbial, especially viral, sequences in cells’ chromosomes is one of the trending theses [17,18].

Some studies found a prominent viral representation within prostate cancer samples, compared with the BPH controls. That includes *Helicobacter pylori*, a human pathogen associated mainly with gastric carcinoma [17]. Nonetheless, it also appears to be a potential risk factor for prostate diseases [28]. With the use of PCR, sequences of *H. pylori* were found integrated at certain locations in the human somatic chromosomes 17, 7, and 11 (17q21.31, 7q21.3, 11q23.2). The integration of the specific cagA gene sequence, which encodes the immune-dominant cagA virulence factor in PPP1R9A and NCAM1 gene locations, could result in dysregulation of their expression. PPP1R9A gene overexpression has been detected in prostate cancer [29] and provides growth advantage to malignant cells. However, the downregulation of NCAM1 genes has been identified in several cancers, suggesting its disrupted repressor function [30]. It is thus assumed that *H. pylori* cagA gene integration may be contributory to overall prostate tumorigenesis. Interestingly, integration of other viral (HPV18, KSHV) sequences was also observed within regions associated with malignant tumor formation (such as FAM111B and KCNC4 genes). Some detected intronic integrations presumably affect splicing during transcription and subsequently alter gene expression [17].

Recent data report that the presence of specific microbes seems to be strongly correlated with biomarkers of prostate cancer. These include increased androgen receptor expression (*Escherichia coli* ETEC H10407 and *Escherichia coli* str. K-12 substr. MG1655), prostate-specific antigen levels (*Campylobacter concisus*, *Streptococcus pneumoniae* SPN032672), Gleason score (*Nevskia ramosa*), stem-cell related gene overexpression (*Staphylococcus aureus* subsp. aureus MW2, *Paraburkholderia phymatum* STM815), and immune-associated gene dysregulation (*Gardnerella vaginalis* 409-05, *Nitrobacter hamburgensis* X14, *Staphylococcus aureus*) [18]. *Gardnerella vaginalis* was found to correlate with the highest number of downregulated immune-associated genes (LPCAT2, TLR3, and TGFB2) and the greatest number of deletions. LPCAT2 was shown to regulate macrophage inflammatory gene expression, along with TLR genes, while TGFB2 protein inhibits oncogenesis by regulating cell growth, proliferation, and apoptosis [31]. Therefore, bacteria could promote prostate tumor progression by actively suppressing immune cell expression, instead of driving inflammation.

Furthermore, some microorganisms were suggested to affect immune processes and contribute to cancer development by active suppression of T-cells’ activity. Microbiome abundance in prostate cancer tissue was indeed highly correlated with expression of regulatory T-cells (Tregs). These cells suppress the activation and proliferation of the effector T-cells, which eventually inhibit a host immune response [18].

## 5. Can Urinary Microbiota Reduce Baseline Risk and Limit Progression?

According to the latest data, microbes may not only play a pro-tumor, but also an anti-tumor role, thus altering the clinical course in prostate cancer patients. Surprisingly, the first study to propose this hypothesis found that most of the microbes detected appeared to have an anti-tumor impact on prostate cancer. These anti-tumor microbes were negatively correlated with cancer phenotypes and could play a significant role in recruiting and boosting immune cells, outcompeting cancer cells, and, overall, inhibiting cancerous process [18]. The mentioned bacteria included *Listeria monocytogenes*, *Methylobacterium radiotolerans* JCM 2831, *Xanthomonas albilineans* GPE PC73, and *Bradyrhizobium japonicum*, which were negatively correlated with Gleason score, tumor–node–metastasis (TNM) stage, prostate-specific antigen (PSA) levels, and androgen receptor (AR) expression, respectively.

Particularly Listeria’s anticancer properties seem of great interest, as it has already been tested in cancer immunotherapy [32]. It secretes protein antigens that stimulate the induction of specific cytotoxic T-cell responses. Pathogen-associated molecular patterns (PAMPs) released from *L. monocytogenes* are recognized by Toll-like receptors (TLRs) present on macrophages and dendritic cells, resulting in immune activation and the secretion of pro-inflammatory cytokines. Apart from triggering a robust immune response, Listeria has also been shown to increase the production of intracellular reactive oxygen species and trigger apoptotic cell death [32].

A separate study stated that *Pseudomonas* infection impedes progression to metastatic disease and might be negatively correlated with TNM as well [19]. The prevalence of *Pseudomonas* in cancer tissues was detected along with the greater expression of small RNAs, in a subset of patients with low rates of metastases [19]. Interestingly, *Stenotrophomonas maltophilia*, which often co-colonizes with *Pseudomonas aeruginosa*, was also found to be negatively correlated with TNM [18].

## 6. Exogenous Microbial Factors—Is STDs’ Role Important?

Recently, probable association between sexual activity and development of genitourinary tumors, including prostate cancer, has been gaining attention. A recent systematic review by Crocetto et al. proclaimed that sexual behaviors, such as number and gender of sexual partners, sexual orientation, ejaculation frequency (EF), and impact of sexually transmitted infections (STIs), may play a role in prostate carcinogenesis. High ejaculatory frequency was considered a protective factor against prostate cancer. Whereas, a high number of sexual partners and sexually transmitted diseases (STDs), especially multiple and lasting episodes, increased the risk of prostate tumor development [33]. Sexually transmitted infections do not patently correlate with increased prostate cancer risk, notwithstanding. Effectively, initial studies did not prove any connection between STIs and higher risk of oncogenesis. Nonetheless, they were small case-control trials with low prevalence of STIs, mainly self-reported by patients [34], which could result in the underestimation of their actual impact. The latest studies consistently state that some patients, previously exposed to STIs, tend to have higher risk of prostate cancer, notwithstanding. Interestingly, these results lead to the assumption that associations between STI history and prostate cancer vary according to ethnicity and indeed may be racial- or ethnic-specific [35,36,37]. A case-control study performed on Mexican men evaluated this assumption and reported that a history of STDs is associated with a nearly two-fold higher risk of prostate cancer development [35]. Another ethnic-stratified analysis proved that solely Latinos reporting any STD and Asian Americans with a history of syphilis and Chlamydia were more likely to develop a prostate cancer. However, due to retrospective design, confounding environmental factors cannot be ruled out. Surprisingly, in both groups, the foreign-born patients experienced a greater risk than those born in the United States [37]. These findings could support the observations that untreated STDs or their delayed treatment in underdeveloped countries may allow for prolonged infection, chronic inflammation, and, ultimately, prostate cancer initiation. This racial-/ethnic specificity, combined with a positive history of STDs, also suggests that both STDs and specific gene variants must be present simultaneously in order to potentially promote prostate carcinogenesis.

Inflammatory environments, driven by sexually transmitted microbes, may modulate innate immune response and its impact is more apparent, when specific gene variants and associated proteins are involved. The following microbes could potentially be more prevalent within certain ethnic groups. Thereby, a proposed combination of factors could eventually lead to enhanced cell proliferation and carcinogenesis. Specifically, pattern recognition receptors, the toll-like receptors (TLR), may be one of the genes in question, as some of their variants have already been linked to an increase or decrease in prostate cancer risk [38,39]. TLRs induce inflammatory cytokine and chemokine genes response, which leads to microbial elimination [40]. Specific prospective data covering prostate cancer development are, however, still lacking.

Despite direct prostate involvement, sexually-transmitted pathogens, not invading the prostatic tissue may still be involved in the development of prostate cancer due to other, yet unknown, mechanisms. Some studies found the positive association between STDs and prostate cancer even more apparent after excluding men with a history of chronic prostatitis [36].

Considering specific sexually transmitted microbial factors, some seem to strongly correlate with promoting progression and invasiveness of prostate cancer in men. This has coherently been shown with regard to *Trichomonas vaginalis* [41,42,43]. Findings suggest that IL-6 and chemokines (such as CCL2 and CXCL8) produced in response to *T. vaginalis* infection induce M2 macrophage polarization and promote neoplastic progression [44]. An extensive review by Crocetto et al. outlined that previous *Trichomonas vaginalis* infection could create a favorable microenvironment, promoting prostate cancer cell proliferation and invasiveness by activating the epithelial–mesenchymal transition (EMT) [45].

Similarly, Chlamydia trachomatis, cultivated within malignant prostate cells line, was found to affect mRNA expression for two major proinflammatory cytokines- IL-6 and FGF-2. They may both account for the mechanism of chemoresistance of prostate cancer, its vascularization, and, thus, the formation of metastatic lesions [46]. Just as promising as these results seem, prospective studies, avoiding self-reported assessments, should be held in order to reduce bias and underestimation of exposure. Voluntary underreport or no knowledge of asymptomatic or mild infections by some microorganisms (*Trichomonas vaginalis*, *Chlamydia trachomatis*) can greatly affect results. Moreover, it is also possible that more prostate cancers are detected in patients with STIs, due to numerous check-ups and the increased rate of imaging. Finally, both multiple episodes of STIs and their duration (as in case of untreated or inefficiently treated infections) may have a diverse impact on prostate carcinogenesis, which is yet to be verified.

## 7. Gut Microbiome and Its Possible Impact on Prostate Carcinogenesis

The fecal microbiome has been associated with risk factors of prostate cancer (obesity, inflammation) and might play a more direct role in carcinogenesis by producing various chemical substances and affecting their metabolism [47]. Some studies revealed a significant difference in microbial diversity between fecal samples from men with and without prostate cancer, with the latter group presenting a higher total count of bacteria, as well as an abundance of rare individual species [48]. It has been thereby postulated that men who are at risk of an aggressive disease would have similarities in their microbial, and thus metabolic, profiles that diverge from those of healthy individuals. On this account, potentially identified microbial differences should not concentrate on the sheer taxonomic composition of the gastrointestinal microbiome, but rather different metabolic and functional profiles, represented by the microbial community [49].

Comparisons of bacterial flora patterns obtained with rectal swabs’ in prostate cancer patients and controls yielded substantial differences. Significant enrichments of *Bacteroides* and *Streptococcus* species were identified in the group with cancer. Moreover, natural B-vitamin production was lacking and the metabolism of both folate and arginine was the most altered in patients with cancer [47]. Interestingly, overall bacteria involved in carbohydrate metabolism pathways were in abundance within the cancer group as well, which is consistent with other studies, reinforcing the thesis [20]. In an attempt to form a novel biomarker of prostate cancer risk, a microbiome-derived risk factor score based on 10 aberrant metabolic pathways was proposed. Unfortunately, the accuracy of the score, when predicting presence of prostate cancer, stayed suboptimal (AUC = 0.64). Therefore, the microbiome score or individual bacterial analysis, for that matter, seems unlikely to become a future biomarker, yet still warrants further investigation. This study also noted higher prevalence of microbiota involved in natural folate production in men without prostate cancer. Therefore, natural folate production could probably be boosted in high-risk men by using probiotics and eliminating external supplementation, as exogenous sources may increase cancer risk [47].

Furthermore, the possibility of an individual’s gastrointestinal microbiome impacting the estrabolome (holistic metabolism of estrogens) may also constitute a mechanism of prostate carcinogenesis. Some epidemiologic and experimental data have pinpointed the estrogens’ impact on prostate cancer initiation and progression [50]. Having analyzed stool samples, another research revealed relevant differences between prostate cancer patients and control cohort, with regard to potential metabolism of estrogens. The higher relative abundance of bacteria-possessing β-glucuronidase genes (*Bacteroides massiliensis*) was seen in the prostate cancer cases, while bacteria lacking these specific genes (*Eubacterium rectale*) were higher in benign controls. β-glucuronidase deconjugation activity leads to higher levels of free estrogens in the bloodstream. These hormones create apurinic sites within DNA that cause mutations, spurring the onset of oncogenesis. The presence of β-glucuronidase activity in cancer patients tends to highlight the importance of the estrabolome theses in the neoplastic process. Nonetheless, despite the auspicious results, *Faecalibacterium* sp., which is positive for β-glucuronidase genes, was also more prevalent in controls, which makes the outcomes vaguer; thus, further insight is required [20].

Finally, there are suggestions that butyrate, an anti-inflammatory micronutrient produced by *Faecalibacterium prausnitzii* and *Eubacterium rectale*, could be implicated in one of the pathways, in turn preventing the development of prostate cancer [45].

Among external factors contributing to dysbiosis, antibiotics are considered to be a major factor disturbing gut microbial diversity, either temporally or permanently. An extensive meta-analysis showed evidence that antibiotic use slightly increases the risk of various cancers, including prostate cancer [51]. Results of statistical comparisons, utilizing preexisting European data, indicated that relatively higher or lower consumption of various antibiotics could be related to certain cancer prevalence figures within European countries. It was reported that countries with high consumption of narrow-spectrum, beta-lactamase-resistant penicillins (Scandinavian countries) showed a higher incidence rate of prostate cancer. Whereas, a negative significance, and a lower incidence rate of prostate cancer, was noted with higher consumption of cephalosporins and quinolones (Slovakia, Greece, Ireland, Czech Republic). It is postulated that altered gut microbiome influences—either promotes or inhibits—carcinogenesis through activating and modulating the host immune response [52]. This cross-sectional data are, however, yet to be validated in well-designed prospective studies.

In contrast to antibiotics, the use of oral probiotics appears to not only improve the intestinal microbial diversity, but also reduce the detrimental effects of long-lasting antibiotic therapy and intestinal dysbiosis. A recent original study by Manfredi et al. suggested that altering gut microbiome with oral administration of probiotic bacterial strain (*Escherichia coli* Nissle 1917- EcN) affects the prostate inflammatory environment. Patients diagnosed with chronic bacterial prostatitis (CBP) were randomized into two groups. All subjects were initially treated with oral levofloxacin, while the experimental group additionally underwent oral administration of EcN. Interestingly, patients on EcN were reported with a significantly lower National Institutes of Health Chronic Prostatitis Symptom Index (NIH-CPSI) score (5.85 ± 3.07 vs. 7.64 ± 3.86; *p* = 0.009) and achieved lower recurrence rate at 3 months (9.8% vs. 26.9%; *p* = 0.043) and at 6 months (8.7% vs. 28.9%; *p* = 0.038). EcN is currently the only known probiotic strain able to compete against *E. coli* involved in the induction of chronic inflammatory environment [53]. Further double-blinded, randomized, controlled trials on different bacterial strains need to be performed. The use of probiotics should also be studied as a potential adjuvant prostate cancer therapy, as it may enhance the chance of responding to the applied treatment and reduce the risk of post-operative infections. This concept seems particularly promising for androgen axis-targeted therapy in prostate cancer treatment, as it was already suggested by Crocetto et al. [45].

Gut microbiota alterations may also be caused by dietary composition, which was extensively described in an up-to-date review covering an interplay between microbiota, prostate cancer, and nutraceutical products. Polyphenol-rich diets or composite polyphenol supplementation were found to increase colonic metabolites, in turn contributing to the chemoprevention of prostate cancer. The intestinal microbiota itself is influenced by the aforementioned colonic metabolites, enabling probiotic bacteria to thrive. Further investigation of the gut microbiome and external factors altering its diversity in context of prostate cancer is essential, in view of possible personalized therapies [45].

## 8. Conclusions

The inapparent role of the microbiome in prostate cancer (Figure 1) is an emerging field that requires further multidisciplinary insight. Both ongoing and future clinical trials shall not only concentrate on detecting microbes, but also explore their complex interactions and metabolic shifts. Certain microbial findings in prostate cancers may prove helpful in predicting clinical diagnosis and potential outcomes of the disease. Their role and a plethora of concomitant connections are estimated to be complex, notwithstanding. As specific microbes may have either positive or negative prognostic and/or diagnostic value (Table 1), manipulating the urinary and gastrointestinal microbiome could eventually improve the patient’s outcome. Exploration of various factors (such as drugs and diet modifications) and their impact on prostate microbiome is required, as they may also be targeted when creating future microbiome-related treatment strategies. Nonetheless, artificial modification of prostate microbiome with microbial transplants or probiotics supplementation could prove beneficial and eventually become vital in holistic prostate cancer patient management.

## Figures and Tables

**Figure 1 jcm-10-04772-f001:**
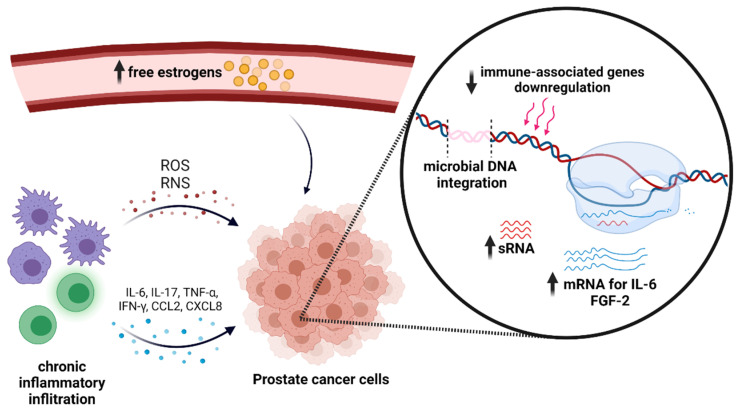
Microbial impact on prostate cancer. Created with BioRender.com (access on 10 July 2021). ROS—reactive oxygen species; RNS—reactive nitrogen species.

**Table 1 jcm-10-04772-t001:** Selected microbial factors and its role in prostate cancer development.

Microbe	Urine/Stool/PC Tissue	Assay	Mechanism of Action	Outcome	Citation
*Propionibacterium* spp.	PC tissue	whole genome sequencing; massive ultradeep pyrosequencing; PCR	Chronic inflammation	Increased prevalence	[21,22,24]
*Staphylococcus* spp.					
*Proteobacterium* spp.					
*Mycoplasma genitalium*					
*Anaerococcus lactolyticus Varibaculum cambriense*	Urine	NGS; PCR	Chronic inflammation	Increased prevalence	[15]
*Actinobaculum schaalii*					
*Propionimicrobium lymphophilum*					
*Ureaplasma* spp.					
*Ochrobactrum* spp.	EPS	PCR-DGGE	Immune dysfunction?	Cancer (stage) progression	[16]
*Helicobacter pylori*	PC tissue	PCR; NGS	cagA integration in PPP1R9A and NCAM	Increased prevalence	[17]
HPV18	PC tissue	PCR; NGS	Viral DNA integration into FAM111B and KCNC4, intergenic and intronic regions	Increased prevalence	[17]
KSHV	PC tissue	PCR; NGS	Viral DNA integration into ORF75, intergenic and intronic regions	Increased prevalence	[17]
*Gardnerella vaginalis*	PC tissue	RNA-sequencing	Downregulation of immune-associated genes (LPCAT2, TLR3, TGFB2)	Increased tumor progression	[18]
*Listeria monocytogenes*	PC tissue	RNA-sequencing	Increased ROS production and cytokines release, induction of cytotoxic T-cells’ response,	Decreased Gleason score	[18]
*Pseudomonas* spp.	PC tissue	DNA and RNA sequencing	Increased small RNAs expression	Reduced progression to metastatic disease	[19]
*Bacteroides massiliensis*	Stool	NGS	Increased free estrogens in bloodstream due to β-glucuronidases	Increased prevalence	[20]
*Trichomonas vaginalis* *	human prostate cancer cell lines (PC3, DU145, and LNCaP)	Quantitative RT PCR, Reverse transcriptase PCR, Western Blot, ELISA Immunofluorescence assay	Increased IL-6 and CCL2 and CXCL8 production; induction of M2 macrophage polarization	Neoplastic progression	[44]
*Chlamydia trachomatis* *	human prostate cancer epithelial cell line (CWR-R1)	Quantitative RT PCR, Immunofluorescence assay	Increased mRNA expression for IL-6 and FGF-2	Chemoresistance and progression to metastatic disease	[46]

*—basic science studies; PC—prostate cancer; PCR—polymerase chain reaction; NGS—next-generation sequencing; EPS—expressed prostatic secretions; RT-PCR—real-time polymerase chain reaction; PCR-DGGE—polymerase chain reaction-denaturing gradient gel electrophoresis.

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
