# Peer review of "The Role of Microbial Factors in Prostate Cancer Development—An Up-to-Date Review"

_jcm, 2021, doi:10.3390/jcm10204772_

Round 1
Reviewer 1 Report
Dear authors
The issue addressed, that of role of the human urinary, intestinal microbiome and sexually-transmitted pathogens in the prostate cancer pathogenesis is very current and challenging. The study is well written, clearly presented and documented, being a review that comprehensively addresses the issues raised.
I just want to point out that table and figure are not mentioned inside the manuscript. So please operate this inclusion.
Author Response
Dear Reviewer,
we are grateful for recognizing our work as interesting. Hereby we provide a point-by-point response to your comments and concerns.
Point 1: Table and figure are not mentioned inside the manuscript.
Response 1: Thank you for pointing this out. We have included citations of the table and the figure in “Conclusions” section of our manuscript.
Point 2: The authors maybe to add a chapter with microbiome diagnostic tools.
Response 2: We have found this suggestion extremely valuable and, accordingly, added a new chapter entitled “Microbiome detection techniques- how did we get here?” to emphasize the complexity and the ongoing development of microbial studies.
Once again, we highly appreciate your valuable feedback on our manuscript.
Reviewer 2 Report
Authors should be congratulated for the great work and the interesting topic proposed. The topic is really challenging and opens to several consideration about the role of pathogens as promoter or starter of the cancer, on sexual transmitted infections (STIs) and the importance (often underestimated) of prevention and the antibiotics abuse in the clinical management of infections. The manuscript is well-written, easily readable, tables and graphics are clearly described, but there are several integrations worthy of discussion:
- Authors must consider Crocetto et al. previous study (PMID: 32878054 DOI: 3390/nu12092648) on crosstalk between Prostate cancer and microbiota inflammation. The manuscript was groundbreaking because it analyzed all the recent evidence on the topic that could be helpful to validate observations presented in the current study.
- Authors should discuss this recent systematic review on the role of sexual activity on the risk of cancer in males (PMID: 34444249 DOI: 3390/ijerph18168500).
- Authors should consider in their review this recent and intriguing manuscript (PMID:34213584DOI: 1007/s00345-021-03773-8). Indeed, the results showed clearly how microbiota influenced prostatic inflammatory environment.
Author Response
Dear Reviewer,
we highly appreciate the thorough and thoughtful comments provided on our submitted article. Attached below are detailed responses to all your suggestions.
Point 1: Authors must consider Crocetto et al. previous study (PMID: 32878054 DOI: 3390/nu12092648) on crosstalk between prostate cancer and microbiota inflammation.
Response 1: Thank you for providing us with this incredibly valuable article. We found it helpful to confront diverse hypotheses and validate our own observations. It also enabled us to cover the role of nutraceutical products, along with their colonic metabolites, on gut microbiome and its combined impact on prostate cancer development. Please find the operated inclusions in our revised manuscript.
Point 2: Authors should discuss this recent systematic review on the role of sexual activity on the risk of cancer in males (PMID: 34444249 DOI: 3390/ijerph18168500).
Response 2: We are grateful for that suggestion. We found this article helpful while discussing the impact of sexually-transmitted diseases on prostate cancer development. Please find this operated inclusion in the chapter “Exogenous microbial factors- is STDs’ role important?”.
Point 3: Authors should consider in their review this recent and intriguing manuscript (PMID:34213584 DOI: 1007/s00345-021-03773-8).
Response 3: Thank you for pointing out this interesting manuscript. It indeed emphasizes the complex impact of gut microbiome on prostate inflammation and puts some light on the role of oral probiotics, as a possible adjuvant therapy in the management of prostate diseases. We have included this article in the revised section “Gut Microbiome and Its Possible Impact on Prostate Carcinogenesis” of our manuscript.
Additionally, we tried our best to improve the language in our manuscript.
We hope that these corrections will meet with approval.
Round 2
Reviewer 2 Report
It's ok.